# Enhancing Robustness in Deep Reinforcement Learning: A Lyapunov Exponent Approach

**Rory Young**[*]     **Nicolas Pugeault**
School of Computing Science
University of Glasgow

## Abstract

Deep reinforcement learning agents achieve state-of-the-art performance in a wide range of simulated control tasks. However, successful applications to real-world problems remain limited. One reason for this dichotomy is because the learnt policies are not robust to observation noise or adversarial attacks. In this paper, we investigate the robustness of deep RL policies to a single small state perturbation in deterministic continuous control tasks. We demonstrate that RL policies can be deterministically chaotic, as small perturbations to the system state have a large impact on subsequent state and reward trajectories. This unstable non-linear behaviour has two consequences: first, inaccuracies in sensor readings, or adversarial attacks, can cause significant performance degradation; second, even policies that show robust performance in terms of rewards may have unpredictable behaviour in practice. These two facets of chaos in RL policies drastically restrict the application of deep RL to real-world problems. To address this issue, we propose an improvement on the successful Dreamer V3 architecture, implementing Maximal Lyapunov Exponent regularisation. This new approach reduces the chaotic state dynamics, rendering the learnt policies more resilient to sensor noise or adversarial attacks and thereby improving the suitability of deep reinforcement learning for real-world applications.

## 1 Introduction

Deep neural networks (DNNs) have revolutionised reinforcement learning (RL) [25], enabling agents to excel in a diverse set of simulated control tasks [12, 16, 22, 23, 24]. However, trained deep RL policies are not robust controllers as DNNs are vulnerable to adversarial attacks [7, 26]. Adding a small amount of noise to each observation can cause these policies to make poor decisions, considerably degrading their overall performance [10, 11, 13]. This lack of stability poses a significant threat when applying deep RL to real-world environments, where inaccurate sensors can easily introduce noise [5]. In this work, we argue that even high-performing deep RL policies are not robust controllers as they can create a chaotic closed-loop control system [4, 14]. These systems are characterised by a high sensitivity to initial conditions, with small changes in initial system states producing vastly different long-term outcomes.

In this paper, we use the spectrum of Lyapunov Exponents [15] to empirically measure the stability of the policies learnt by state-of-the-art deep RL approaches subject to small state perturbations. We show that these controllers can produce chaotic state and reward dynamics when controlling continuous environments. Consequently, a single noisy observation has a dramatic long-term impact on these control systems, with the subsequent approximate state and reward trajectories diverging significantly (Figure 1). This instability poses two problems for the safe deployment of deep RL in real-world environments.

---

[*]Corresponding email: R.Young.4@research.gla.ac.uk

38th Conference on Neural Information Processing Systems (NeurIPS 2024).

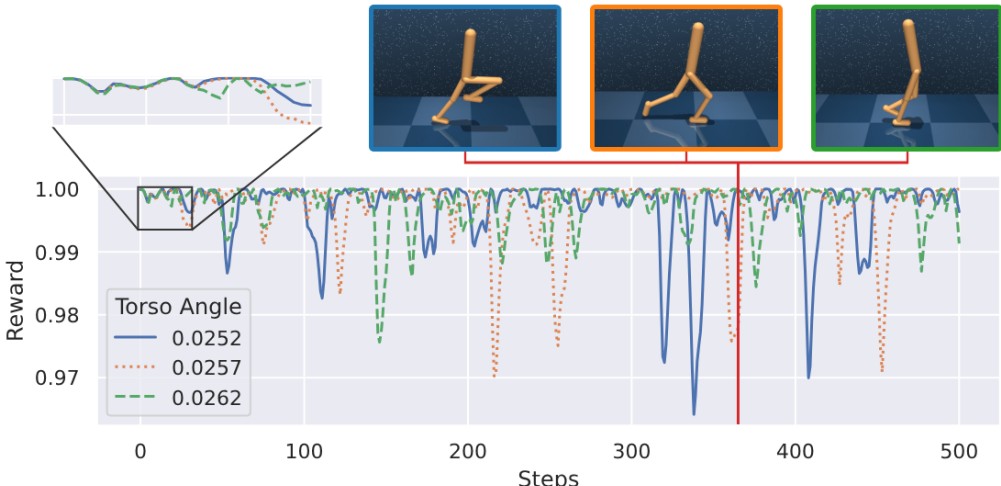

Figure 1: Reward attained when a trained deterministic Soft Actor-Critic [8] agent controls the deterministic *Walker Walk* environment. Each system has the same initial configuration other than the torso angle, which is perturbed by $\pm 5 \times 10^{-4}$ degrees. This small perturbation causes the systems to significantly diverge after 50 steps due to the chaotic nature of the control interaction. Consequently, this affects overall performance as there is significant variation in the total reward attained.

1. The chaotic state dynamics create a fractal return surface [28] which is highly sensitive to small changes in the system state. The high-frequency oscillations in this function cause a lack of robustness as small state perturbations can produce significantly different total rewards.

2. Even for high-performing policies with stable returns, it remains impossible to accurately predict the long-term behaviour of these chaotic control systems as they rely on noisy partially observable sensors to attain an observation. This unpredictability means that safe and reliable behaviour cannot be guaranteed.

To address these issues, we propose Maximal Lyapunov Exponent regularisation for Dreamer V3 [9]. This novel technique estimates the local state divergence using the Recurrent State Space Model and incorporates this term into the policy loss. We demonstrate that this regularisation term significantly reduces the chaotic dynamics produced by this state-of-the-art deep RL controller. This increased stability dramatically improves the robustness of the policy, thus improving the feasibility of deep RL agents for real-world continuous control tasks.

## 2 Background

### 2.1 Robust reinforcement learning

Reinforcement learning provides a data-driven method for solving sequential decision-making problems. This control interaction is represented by a deterministic Markov Decision Process (MDP) with state space $\mathcal{S} \subseteq \mathbb{R}^n$, action space $\mathcal{A} \subseteq \mathbb{R}^m$, scalar reward function $r : \mathcal{S} \times \mathcal{A} \to \mathbb{R}$, state transition function $f : \mathcal{S} \times \mathcal{A} \to \mathcal{S}$ and initial state distribution $\rho_0 \subseteq \mathcal{S}$. The objective of an RL agent is to learn a policy $\pi_\theta : \mathcal{S} \to \mathcal{A}$ which maximises the sum of discounted returns (Equation 1) for a given discount factor $\gamma \in [0, 1)$.

$$J(\theta) = \mathbb{E}_{s_0 \sim \rho_0} \left[ \sum_{t=0}^{\infty} \gamma^t \times r(s_t, a_t) \,\middle|\, s_{t+1} = f(s_t, a_t),\ a_t = \pi_\theta(s_t) \right] \tag{1}$$

RL policies are said to be robust if they can maintain consistent behaviour and reliable performance in the face of noise or adversarial attacks. This stability is crucial for the safe deployment of deep RL

Table 1: Stability of a dynamical system for different values of the Maximal Lyapunov Exponent ($\lambda_1$) and Sum of Lyapunov Exponents ($\lambda_\Sigma$).

| $\lambda_1$ | $\lambda_\Sigma$ | Stability |
|:---:|:---:|:---|
| - | - | Stable |
| + | - | Chaotic |
| + | + | Unstable |

to real-world applications where observation noise is inevitable. Recently, a wide range of attack methods have been developed which can easily compromise the performance of deep RL policies with relatively low levels of intervention. In their seminal work, Huang et al. [10] extended the Fast Gradient Sign Method [7] attack to deep RL policies, demonstrating that the addition of a small amount of strategic noise to every observation is sufficient to degrade the performance of trained deep RL agents. Furthermore, Kos & Song [11] showed that this attack does not need to occur at every step, as less frequent attacks with small Gaussian noise still produce a drop in performance. Additionally, they demonstrated that only perturbing the observation when the value function surpasses a set threshold still produces poor performance while significantly decreasing the frequency of the attack. A similar result was established by Lin et al. [13], who showed that specifically attacking when the relative performance gain between best and worst action surpasses a defined threshold can successfully degrade performance. These findings demonstrate that consistent and inconsistent small perturbations to the system can easily confuse deep RL policies, resulting in unintended and detrimental behaviour. Note that small observation noise is frequent in real-world systems, and therefore, this lack of robustness severely limits the application of deep RL to real-world environments.

## 2.2 Measuring stability: Lyapunov Exponents

Lyapunov Exponents [15, 20] provide a method for quantifying the stability of complex, non-linear, high-dimensional systems by measuring the deformation rate of a small hyperellipsoid under the effects of a transition function. In general, for a dynamical system with $N$ degrees of freedom, there are $N$ Lyapunov Exponents, each representing the exponential growth rate of a unique principal axis of the hyperellipsoid. Given a set of $N$ ordered exponents ($\lambda_1 \geq \lambda_2 \geq ... \geq \lambda_N$), the volume of the hyperellipsoid grows proportionally to $e^{(\lambda_1+\lambda_2+...\lambda_N)t}$ and the length grows proportionally to $e^{\lambda_1 t}$. From this definition, the Maximal Lyapunov Exponent (MLE) (Equation 2) and the Sum of Lyapunov Exponents (SLE) (Equation 3) are used to determine if a system is stable, chaotic or unstable, as outlined in Table 1.

$$\lambda_1 = \lim_{t \to \infty} \lim_{\hat{s}_0 \to s_0} \frac{1}{t} \ln\left(\frac{|s_t - \hat{s}_t|}{|s_0 - \hat{s}_0|}\right) \quad (2) \qquad\qquad \lambda_\Sigma = \sum_{i=0}^{N} \lambda_i \quad (3)$$

Dynamical systems with a negative MLE ($\lambda_1 \leq 0$) are stable as all principal axes of the hyperellipsoid exponentially decrease to zero [21]. In these systems, any trajectories produced from similar initial positions converge to the same trajectory given sufficient time. Conversely, systems with positive MLE ($\lambda_1 > 0$) and SLE ($\lambda_\Sigma > 0$) are unstable as the resulting state trajectories diverge at an exponential rate. However, for a positive MLE ($\lambda_1 > 0$) and negative SLE ($\lambda_\Sigma < 0$), similar trajectories will diverge at an exponential rate but remain confined to a subregion of the phase space known as a *chaotic attractor* [4, 14]. This bounded exponential divergence means trajectories in this region of the state space are unstable and only replicable given the exact same starting state: small perturbations to the starting state will produce significantly different long-term outcomes which appear random and uncorrelated. As a result, *it is impossible to predict the long-term behaviour of a chaotic system given an approximation of the initial state*.

To measure the stability of a known dynamical system, the full spectrum of Lyapunov Exponents can be estimated using the approach outlined by Benettin et al. [2, 3]. This method represents the spectrum as a set of small perturbation vectors which are iteratively updated using the known transition function. To avoid all vectors collapsing in the direction of maximal growth, they are periodically

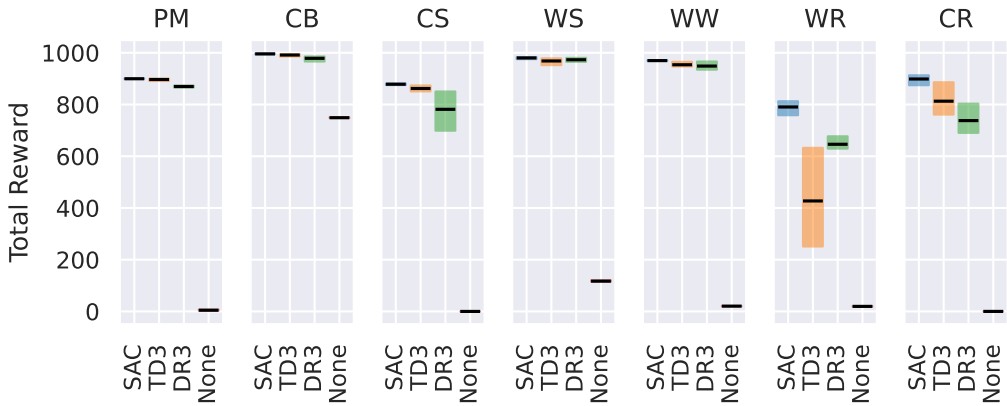

Figure 2: Total episode reward for the *Pointmass Easy* (PM), *Cartpole Balance* (CB), *Cartpole Swingup* (CS), *Walker Stand* (WS), *Walker Walk* (WW), *Walker Run* (WR) and *Cheetah Run* (CR) environments when controlled by trained instances of SAC, TD3, Dreamer V3 (DR3) and an agent which takes no actions (None). Each policy-environment combination is independently trained with three random seeds and the average interquartile episode reward with a bootstrapped 95% confidence interval is reported over 80 evaluation episodes each with a fixed length of 1000 steps.

Gram-Schmidt orthonormalised so that each vector maintains a unique direction. Performing this orthonormalisation allows for the detection of both positive and negative Lyapunov exponents up to the dimension of the phase space. The spectrum of Lyapunov Exponents is then determined as the average log rate of divergence of the perturbation vectors as they are Gram-Schmidt orthonormalised. Estimation of the full spectrum of Lyapunov Exponents using this method allows for the estimation of $\lambda_1$ and $\lambda_\Sigma$ which can quantify the stability of the dynamical system.

### 2.3 Chaos in reinforcement learning

Previous studies have investigated the chaotic state and reward dynamics produced by reinforcement learning policies. Rahn et al. [19] showed that the policy optimisation landscape can contain high-frequency discontinuities in the vicinity of a trained policy. A similar result was established by Wang et al. [28], who proved that control systems with Lipschitz continuous reward and transition functions only have a Lipschitz continuous objective function if $\lambda_1 < -\ln(\gamma)$. When $\lambda_1 > -\ln(\gamma)$, the objective function is a fractal and is $\alpha$-Hölder continuous with holder exponent $\alpha = -\ln(\gamma)/\lambda_1$. Consequently, a single update to the policy in these chaotic control systems can produce substantially different total rewards. Furthermore, Parmas et al. [17] demonstrated that chaos exists in model-based RL methods due to repeated nonlinear predictions and this instability causes gradients to explode during training.

While these works use chaos theory to highlight an important issue in the field of RL policy learning, they are focused primarily on the stability of the policy subject to *policy parameter* perturbations during training. Our work uses similar concepts but instead focuses on the stability of fully trained RL policies subject to *state perturbations* and the adverse effect this has on the total reward attained in realistic environments. To our knowledge, we are the first to use the spectrum of Lyapunov Exponents to estimate the level of chaos produced by trained DNN policies in continuous control tasks.

## 3 Chaotic state dynamics

In this section, we use Lyapunov Exponents to identify the level of chaos produced by various state-of-the-art deep reinforcement learning policies in continuous control environments. We claim that the presence of chaos in the MDP implies that the policies are not robust controllers, as trivial changes to the system state produce significantly different long-term state trajectories. This instability poses a significant problem for real-world control systems where consistent and predictable behaviour is necessary.

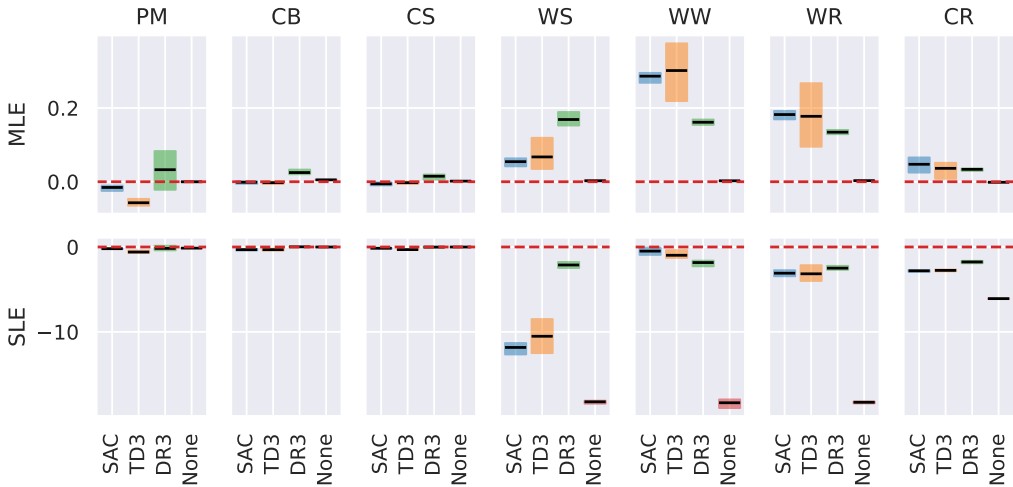

Figure 3: Estimated Maximal Lyapunov Exponent (MLE) and Sum of Lyapunov Exponents (SLE) for the *Pointmass* (PM), *Cartpole Balance* (CB), *Cartpole Swingup* (CS), *Walker Stand* (WS), *Walker Walk* (WW), *Walker Run* (WR) and *Cheetah Run* (CR) environments when controlled by a trained instance of SAC, TD3, Dreamer V3 (DR3) and an agent which takes no actions (None). Each policy-environment combination is independently trained with three random seeds and the interquartile average MLE & SLE for each seed is calculated using 20 initial states. A bootstrapped 95% confidence interval is included to show the variation in MLE and SLE across random seeds.

To closely match real-world applications, we estimate the stability of tasks from the DeepMind Control Suite [27] when controlled by deep RL policies. These simulated environments provide a range of deterministic control problems with continuous state spaces, as outlined in Appendix A.1. For each control task we independently train three instances of Soft Actor-Critic (SAC) [8], Twin Delayed Deep Deterministic Policy Gradients (TD3) [6] and Dreamer V3 [9], as these represent state-of-the-art off-policy and model-based methods for continuous control tasks. Furthermore, to determine if trained deep RL policies directly influence the stability of each control system, we also introduce a passive controller that takes no actions. All models are based on the Stable Baselines 3 [18] implementation with the parameters outlined in Appendix A.2 and trained using an Intel Core i7-8700 CPU workstation with an Nvidia RTX 2080 Ti GPU and 32GB of RAM. The average interquartile reward and bootstrapped 95% confidence interval [1] for each policy type are reported in Figure 2. These results show that all three algorithms learn policies which performed well and significantly better than the no-action baseline. However, this does not speak to the types of dynamics produced or how robust these policies are to local state perturbations.

To identify the stability of each policy-environment interaction, we estimate the full spectrum of Lyapunov Exponents using the method proposed by Benettin et al. [2, 3]. The spectrum is calculated using states sampled from the initial state distribution for each environment. A perturbation vector is initialised for each state dimension at a distance of $10^{-4}$ from the sample state. This represents an arbitrarily small change to the control system without introducing numerical precision errors. The spectrum is calculated over 1000 environment steps and perturbation vectors are orthonormalised every 10 steps to prevent divergence saturation. This process is repeated with 20 initial states and the average value for each exponent is used to calculate the MLE and SLE. Ablation studies for these constants are provided in Appendix B.

Figure 3 provides the bootstrapped 95% confidence interval for the MLE and SLE produced by each policy-environment pair. These results indicate that all the environments are naturally invariant to small perturbations as the no action baseline has $\lambda_1 = 0$. Conversely, when these environments are controlled by deep RL policies, $\lambda_1$ can be non-zero, indicating that DNN controllers directly influence the stability of these control systems.

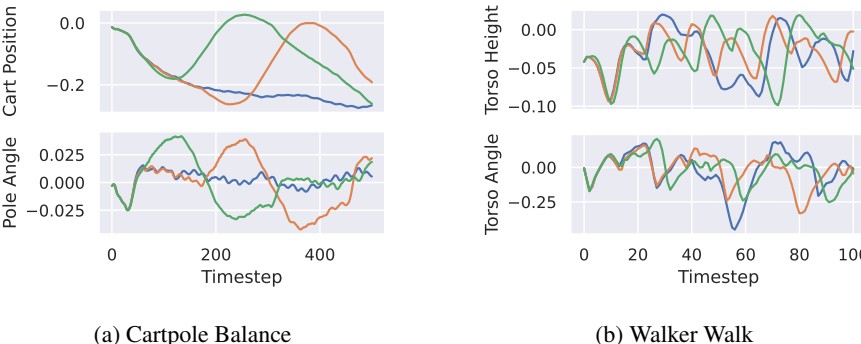

(a) Cartpole Balance                    (b) Walker Walk

Figure 4: Partial state trajectory produced by Dreamer V3 when controlling *Cartpole Balance* and *Walker Walk* subject to a single initial state perturbation. Initially, each system is separated by only $10^{-4}$ units but the subsequent state trajectories diverge significantly as the control interaction is chaotic.

We find that *SAC* and *TD3* produce stable state dynamics in simple low-dimensional environments (*Pointmass*, *Cartpole Balance* and *Cartpole Swingup*) as they have negative MLE. Therefore, if a small perturbation is made to any of the states in these environments, the subsequent state trajectories converge. This result is consistent with the definition of the reward function for each of these tasks as they provide high rewards for maintaining the system at a fixed location. However, when controlled by Dreamer V3, these simple systems exhibit low levels of chaos as they have positive MLE and negative SLE. As a result, state trajectories in these environments are highly sensitive to initial conditions, with similar states producing significantly different long-term outcomes as shown in Figure 4a. Instead of converging to a single state, these trajectories exhibit chaotic behaviour, continuously orbiting within a region which yields high rewards.

Furthermore, all deep RL methods produce chaotic dynamics in the complex high-dimensional environments (*Walker Stand*, *Walker Walk*, *Walker Run* and *Cheetah Run*), as indicated by the positive MLE and negative SLE. This means that *these policies cannot account for arbitrarily small changes* in the system's state since small changes produce exponentially diverging state trajectories, as illustrated in Figure 4b. This poses a significant problem for real-world applications of RL, as observation perturbations are easily introduced via imperfect measurements or sensor noise. Therefore, for these complex high-dimensional environments controlled by deep RL policies, it is impossible to guarantee stability as the system cannot correct itself after a single inaccurate observation.

## 4   Chaotic rewards

In this section, we show that chaotic state trajectories can also impact a policy's performance and produce chaotic reward trajectories as determined by the Maximal Lyapunov Exponent. This instability creates a fractal return surface in which small state perturbations produce significantly different total rewards. We argue that adversarial attack methods could leverage these high-frequency oscillations, repeatedly injecting perturbations which cause the agent to follow the state trajectories that attain the lowest total reward. This lack of robustness poses a significant problem for real-world control systems, where the worst-case performance is often more significant than the average performance.

By considering the reward over a state trajectory as a trajectory in a one-dimensional reward space, the stability of the reward can also be measured using Lyapunov Exponents. Given that this space is one-dimensional, only one Lyapunov Exponent exists; however, this single exponent can still be used to reliably identify the stability of reward trajectories. Negative $\lambda_1$ values indicate that small perturbations to the system's state still produce converging long-term rewards. Moreover, for a bounded reward function, the reward trajectories cannot diverge indefinitely; thus a positive $\lambda_1$ indicates that the long-term reward is chaotic as small changes to the system state produce exponentially diverging bounded reward trajectories.

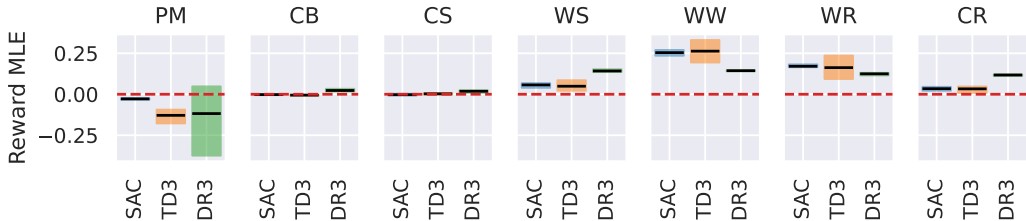

Figure 5: Reward MLE interquartile mean for the *Pointmass* (PM), *Cartpole Balance* (CB), *Cartpole Swingup* (CS), *Walker Stand* (WS), *Walker Walk* (WW), *Walker Run* (WR) and *Cheetah Run* (CR) when controlled by SAC, TD3 and Dreamer V3 (DR3). Each policy-environment combination is independently trained with three random seeds and the reward MLE for each seed is calculated using 20 initial states. A bootstrapped 95% confidence interval is included to show the variation in reward stability across random seeds.

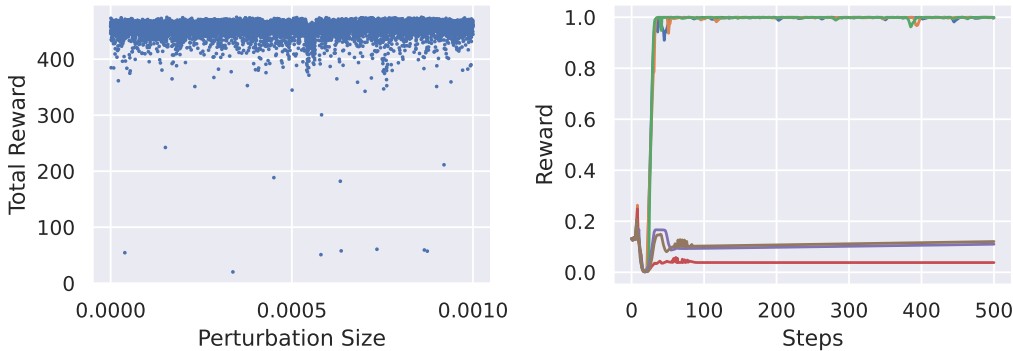

Figure 6: Left: Total reward attained by a deterministic SAC policy when controlling the deterministic *Walker Walk* environment subject to an initial perturbation with fixed direction and varying magnitude. Right: Rewards attained by the three best and three worst state trajectories subject to this perturbation.

Figure 5 provides the average reward MLE and bootstrapped 95% confidence interval for each policy-environment pair defined in Section 3. These graphs indicate that the reward trajectories produced in simple low-dimensional environments (*Pointmass*, *Cartpole Balance*, *Cartpole Swingup*) are stable as they have negative reward MLE. As a result, a single small state perturbation does not harm the total reward attained as the subsequent reward trajectories converge. Conversely, in high-dimensional control systems (*Walker Stand*, *Walker Walk*, *Walker Run* and *Cheetah Run*), the reward MLE is positive, indicating that the reward is highly sensitive to arbitrarily small changes in the system state. Consequently, reward trajectories in these systems are unstable, as similar initial states produce significantly different sequences of rewards. Plotting the reward trajectories for one of these chaotic interactions (Figure 6) shows that a single perturbation with varying magnitude can have a huge detrimental impact on the total reward attained *despite performing well on average*. This poses a significant problem for real-world applications of RL as it is impossible to guarantee worst-case performance in a chaotic control system.

## 5 Maximal Lyapunov Exponent regularisation

In Sections 3 & 4, we established that deep RL policies can produce chaotic state trajectories in continuous control tasks and that this can have a large detrimental impact on performance. To address this, we propose a novel regularisation method which improves the stability of RL policies by constraining the Maximal Lyapunov Exponent during policy updates. This improved stability is a crucial step towards the safe and reliable deployment of RL policies in real-world domains where local perturbations are common.

---

**Algorithm 1** MLE regularisation

---

**Require:**
  Policy ($\pi_\theta : \mathcal{H} \times \mathcal{Z} \to \mathcal{P}(\mathcal{A})$)
  Encoder ($q_\phi : \mathcal{S} \times \mathcal{H} \to \mathcal{P}(\mathcal{Z})$)
  Decoder ($p_\phi : \mathcal{H} \times \mathcal{Z} \to \mathcal{P}(\mathcal{S})$)
  Dynamics Predictor ($q_\phi : \mathcal{H} \to \mathcal{P}(\mathcal{Z})$)
  Sequence Model ($f_\phi : \mathcal{H} \times \mathcal{Z} \times \mathcal{A} \to \mathcal{H}$)

  Current State ($s \in \mathcal{S}$)
  Current Hidden State ($h \in \mathcal{H}$)
  Time Horizon ($T \in \mathbb{N}$)

**Ensure:** $\mathcal{L}^{\lambda_1}(\theta)$
  $\mathcal{L}^{\lambda_1}(\theta) \leftarrow 0$        # Initialise the MLE regularisation loss
  $Z = \{z_l \sim q_\phi(s, h)\}_{l=1}^{L}$        # Update stochastic representation
  **for** $t = 1, 2, ..., T$ **do**
    $A \leftarrow \{a_l \sim \pi_\theta(H_l, Z_l)\}_{l=1}^{L}$        # Generate a set of sample action
    $H \leftarrow \{h_l = f_\phi(H_l, Z_l, A_l)\}_{l=1}^{L}$        # Update hidden representation
    $Z \leftarrow \{z_l \sim q_\phi(H_l)\}_{l=1}^{L}$        # Update stochastic representation
    $S \leftarrow \{s_l \sim p_\phi(H_l, Z_l)\}_{l=1}^{L}$        # Generate a set of predicted states
    $\mathcal{L}^{\lambda_1}(\theta) \leftarrow \mathcal{L}^{\lambda_1}(\theta) + \mathrm{Var}(S) + \mathrm{Var}(H)$        # Update the MLE regularisation loss
  **end for**
  **return** $\mathcal{L}^{\lambda_1}(\theta)$

---

We base our regularisation on Dreamer V3 [9], a general-purpose model-based RL algorithm which attains state-of-the-art performance across a diverse set of control tasks. To achieve this, Dreamer V3 uses a Recurrent State Space Model (RSSM) consisting of an Encoder ($q_\phi : \mathcal{S} \times \mathcal{H} \to \mathcal{P}(\mathcal{Z})$), Decoder ($p_\phi : \mathcal{H} \times \mathcal{Z} \to \mathcal{P}(\mathcal{S})$), Dynamics Predictor ($q_\phi : \mathcal{H} \to \mathcal{P}(\mathcal{Z})$) and Sequence Model ($f_\phi : \mathcal{H} \times \mathcal{Z} \times \mathcal{A} \to \mathcal{H}$) to predict state trajectories ($s_t$), bootstrapped $\lambda$-return trajectories ($R_t^\lambda$) [25] and state value trajectories ($v_\phi(s_t)$) over a short time horizon $T$. The policy is then trained to maximise the normalised advantage estimates using REINFORCE gradients [29] and an entropy regulariser (H[·]) [30] with weighting coefficient $\eta$. The full loss function used to train Dreamer V3's policy is outlined in Equation 4.

$$\mathcal{L}^{\mathrm{Dr3}}(\theta) \doteq -\sum_{t=1}^{T} \left[ \mathrm{sg}\left( \frac{R_t^\lambda - v_\phi(s_t)}{\max(1, S)} \right) \log \pi_\theta(a_t | s_t) + \eta \mathrm{H}\left[ \pi_\theta(a_t | s_t) \right] \right] \tag{4}$$

$$\mathcal{L}^{\lambda_1}(\theta) \doteq \sum_{t=1}^{T} \left[ \mathop{\mathrm{Var}}_{L}(S_t) + \mathop{\mathrm{Var}}_{L}(H_t) \right] \tag{5}$$

$$\mathcal{L}^{\mathrm{Policy}}(\theta) \doteq \mathcal{L}^{\mathrm{Dr3}}(\theta) + \mathcal{L}^{\lambda_1}(\theta) \tag{6}$$

At its core, Dreamer V3 uses a stochastic RSSM to predict the state and reward trajectories over a predefined time horizon given an initial starting state $s_0$ and an internal representation $h_0$. Due to the stochastic nature of this model, repeating the same trajectory predictions $L \in \mathbb{N}$ times produces a set of state trajectories ($S_t = \langle s_{t,1}, s_{t,2}, ..., s_{t,L} \rangle$) and internal representation trajectories ($H_t = \langle h_{t,1}, h_{t,2}, ..., h_{t,L} \rangle$), each of which provides a plausible estimate of the future states. The variance between trajectories ($\mathrm{Var}_L(\cdot)$) thus provides an estimation of the local state divergence as the state perturbation size approaches 0. Therefore, to minimise $\lambda_1$ and improve the stability of Dreamer V3 subject to state perturbation, we propose incorporating the regularisation term outlined in Equation 5 into the policy loss (Equation 6). Including this regularisation term as an additional weighted term forces agents to consider the stability of the system during the optimisation process. This incentivises the policy to produce stable state trajectories which attain high rewards instead of solely optimising the expected return. The complete algorithm for calculating the MLE regularisation term is provided in Algorithm 1 using the notation outlined by Hafner et al. [9].

Table 2: Average total reward and average MLE produced when Dreamer V3 (DR3) and Dreamer V3 with MLE regularisation (MLE DR3) when controlling various environments sampled from the *DeepMind Control Suite* [27]. Each policy-environment combination is independently trained with three random seeds using the hyperparameters outlined in Appendix A.2.

| | Reward | | MLE | |
| Environment | DR3 | MLE DR3 | DR3 | MLE DR3 |
|---|---|---|---|---|
| Pointmass | 869.5 | **880.5** | 0.0326 | **-0.0275** |
| Cartpole Balance | **978.6** | 970.5 | 0.0249 | **0.0231** |
| Cartpole Swingup | 781.4 | **866.4** | **0.0149** | 0.0235 |
| Walker Stand | **973.0** | 961.6 | 0.1688 | **0.0654** |
| Walker Walk | 948.6 | **950.7** | 0.1614 | **0.1405** |
| Walker Run | 646.3 | **698.4** | 0.1345 | **0.1106** |
| Cheetah Run | **737.7** | 675.2 | 0.0337 | **0.0283** |

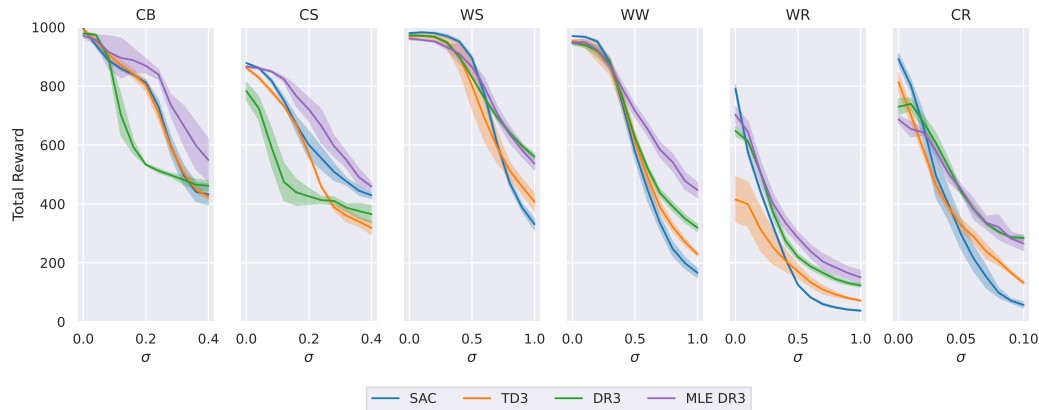

Figure 7: Total episode reward for the *Cartpole Balance* (CB), *Cartpole Swingup* (CS), *Walker Stand* (WS), *Walker Walk* (WW), *Walker Run* (WR) and *Cheetah Run* (CR) environments when controlled by trained instances of SAC, TD3, Dreamer V3 (DR3) and Dreamer V3 with MLE regularisation (MLE DR3) subject to $\mathcal{N}(0, \sigma)$ Gaussian observation noise. Each policy-environment combination is independently trained with three random seeds and the average episode reward with a bootstrapped 95% confidence interval is reported over 80 evaluation episodes each with a fixed length of 1000 steps.

## 6   Experiments

In this section, we investigate the impact that the proposed MLE regularisation has on Dreamer V3. We show that the inclusion of this term reduces the chaotic state dynamics produced by the control policy and that this improved stability increases performance when noise is introduced. For these experiments, we train three instances of Dreamer V3 with MLE regularisation and reuse the SAC, TD3 and Dreamer V3 policies from Sections 3 & 4. When estimating state divergence, the RSSM predicts $L = 3$ plausible future trajectories over which the state and internal representation variance is measured. Increasing $L$ will produce a more accurate estimate of state divergence; however, this will require more computational resources. Therefore, to maintain a similar training time to that of Dreamer V3, we set $L = 3$. All other hyperparameters are consistent with the Dreamer V3 baseline.

Table 2 provides the average reward and estimated MLE produced by the regularised and unregularised Dreamer V3 models for each control task. This indicates that MLE regularisation successfully minimises the chaotic state dynamics produced by Dreamer V3 while maintaining similar performance. However, as MLE is still positive in the majority of environments, the control interaction still produced chaotic state trajectories. Despite this, the regularised policies are more stable as the rate of divergence has significantly decreased.

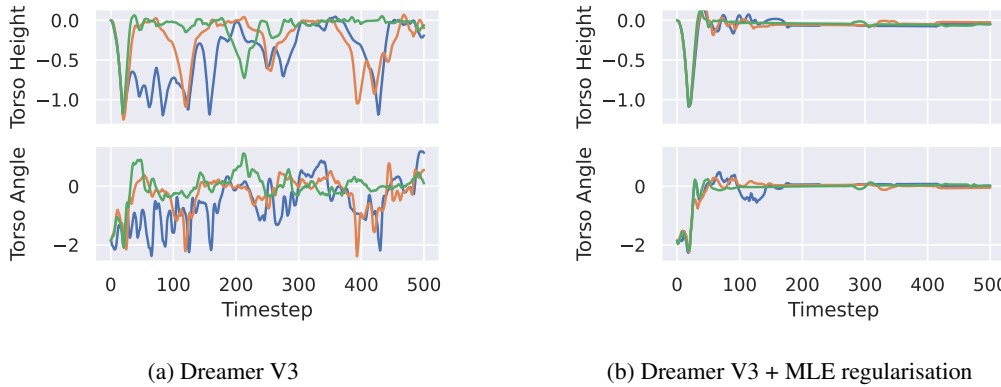

(a) Dreamer V3                                    (b) Dreamer V3 + MLE regularisation

Figure 8: State trajectories produced when Dreamer V3 and Dreamer V3 + MLE regularisation control the *Walker Stand* environment with $\mathcal{N}(0, 0.5)$ Gaussian observation noise.

To identify how robust these regularised Dreamer V3 policies are to continual state perturbations, we measure their performance when Gaussian noise is added to each observation. Each policy is trained without observation noise ($\sigma = 0$) in the fully deterministic variant of each environment and then tested with Gaussian noise ($\mu = 0$ and $\sigma \in [0, 1]$). Performance is measured over 240 episodes with a fixed length of 1000 steps and a maximum reward per step of 1. Figure 7 shows the interquartile mean and bootstrapped 95% confidence interval for the total reward attained by each policy-environment interaction subject to various levels of Gaussian noise. This shows that MLE regularisation significantly improves the performance of Dreamer V3 in four of the noisy control systems, while the other two environments (*Walker Stand* and *Cheetah Run*) attain similar performance to the Dreamer V3 baseline. Furthermore, examining the state trajectories produced by Dreamer V3 and Dreamer V3 + MLE regularisation when controlling the *Walker Stand* task (Figure 8) shows that the regularisation improves the stability of the control interaction as the trajectories do not diverge significantly. These findings indicate that MLE regularisation can improve the robustness of Dreamer V3 subject to observation noise as it produces more consistent and reliable behaviour in the face of uncertainties.

## 7    Conclusion

A key issue preventing the application of deep reinforcement learning to real-world environments is the need for guaranteed stability and performance in the face of noisy observations and adversarial attacks. In this work, we set out to identify the impact a single perturbation has on the long-term behaviour of deep RL policies in continuous control environments. Using the spectrum of Lyapunov Exponents, we established that the MDP can produce chaotic state and reward trajectories which are highly sensitive to initial conditions. This instability poses two threats to the application of deep RL to real-world problems, where it is infeasible to attain an accurate measurement of the system state. First, small state perturbation can have a large impact on the performance of trained deep RL policies, even where the average performance is good. This can create hazards in real-world conditions where observation noise is prevalent and can be exploited by adversarial attack methods. Second, even when deep RL policies perform well, they can produce unpredictable behaviours, which is undesirable in most real-world applications.

To mitigate these chaotic dynamics and improve robustness we propose Maximal Lyapunov Exponent regularisation for Dreamer V3. This novel approach uses the Recurrent State Space Model to estimate the local state divergence and incorporates this into the policy loss. In effect, the agent optimises its confidence in future trajectories jointly with its expectations of rewards. While MLE regularisation helps improve the robustness of the agent's policies, this approach assumes an accurate estimation of the local state divergence. In environments where the RSSM struggles to capture state dynamics, the effectiveness of the proposed regularisation may be diminished. However, in our experiments, we demonstrate that this regularisation improves the stability of the learnt policies, thereby making them more robust to state perturbations.

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

# A  Appendix

## A.1  State space composition

Table 3: State space definition for each environment used in this paper. Each dimension of the state space represents a unique aspect of the control system and can contain any real value. Positions are measured in metres ($m$), angles are measured in radians ($rad$), velocities are measured in metres per second ($m/s$) and angular velocities are measured in radians per second ($rad/s$).

| Render | Name | Degrees of freedom | Axis Representations |
|---|---|---|---|
|  | Pointmass | 4 | Point mass x & y position
Point mass x & y velocity |
|  | Cartpole Balance
Cartpole Swingup | 4 | Cart position
Cart velocity
Pole angle
Pole angular velocity |
|  | Walker Stand
Walker Walk
Walker Run | 18 | Torso x & z position
Torso x & z velosity
Torso angle
Torso angular velocity
Left & right hip angle
Left & right hip angular velocity
Left & right knee angle
Left & right knee angular velocity
Left & right ankle angle
Left & right ankle angular velocity |
|  | Cheetah Run | 18 | Torso x & z position
Torso x & z velocity
Torso angle
Torso angular velocity
Front & back hip angle
Front & back hip angular velocity
Front & back knee angle
Front & back knee angular velocity
Front & back ankle angle
Front & back ankle angular velocity |

## A.2 Hyperparameters

Table 4: Hyperparameters used to train SAC, TD3 and Dreamer V3

**SAC**

| Parameter | Value |
| --- | --- |
| Environment steps | $5 \times 10^6$ |
| Buffer size | $10^6$ |
| Parallel environments | 8 |
| Update period | 8 |
| Updates per step | 8 |
| Discount factor | 0.99 |
| Learning rate | 0.0003 |
| Batch size | 1024 |
| Polyak update coefficient | 0.005 |
| Networks activation | Tanh |
| Networks depth | 3 |
| Networks width | 256 |

**TD3**

| Parameter | Value |
| --- | --- |
| Environment steps | $5 \times 10^6$ |
| Buffer size | $10^6$ |
| Parallel environments | 8 |
| Update period | 8 |
| Updates per step | 8 |
| Discount factor | 0.99 |
| Learning rate | 0.0003 |
| Batch size | 1024 |
| Polyak update coefficient | 0.005 |
| Networks activation | Tanh |
| Networks depth | 3 |
| Networks width | 256 |

**Dreamer V3**

| Parameter | Value |
| --- | --- |
| Environment steps | $10^6$ |
| Buffer size | $10^5$ |
| Parallel environments | 8 |
| Update period | 8 |
| Updates per step | 8 |
| Discount factor | 0.99 |
| Learning rate | $10^{-4}$ |
| Batch size | 15 |
| RSSM batch length | 64 |
| Imagination horizon | 15 |
| Network activation | LayerNorm + SiLU |
| Networks depth | 2 |
| Networks width | 512 |
| Recurrent state size | 4096 |
| Number of latents | 32 |
| Classes per latent | 32 |

# B  Lyapunov Exponent ablation study

## B.1  Default values

Table 5: Parameters used to calculate the spectrum of Lyapunov Exponents using the method outlined by Benettin et al. [2, 3].

| Parameter | Value |
| --- | --- |
| Total timesteps | 1000 |
| Number of iterations | 100 |
| Normalisation period | 10 |
| Number of samples | 20 |
| Perturbation size | 0.0001 |

## B.2 Number of iterations ablation study

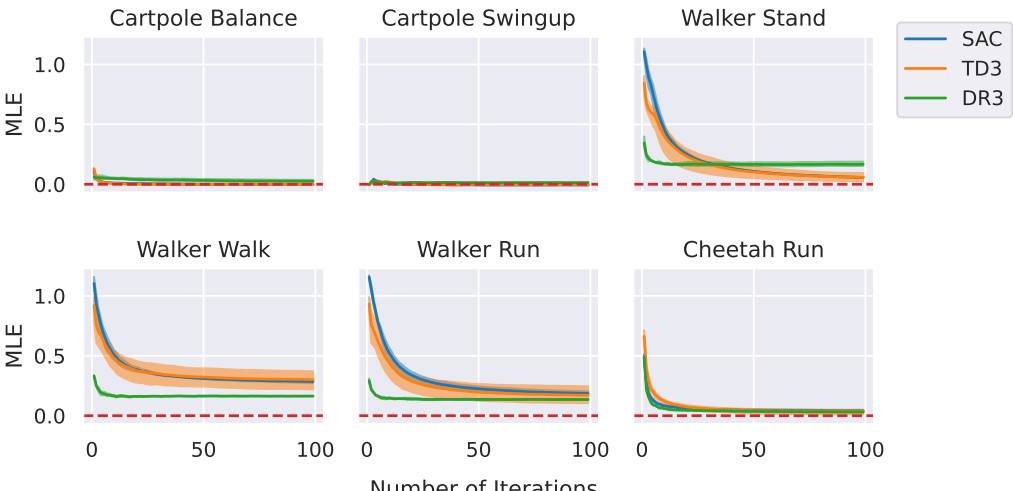

Figure 9: Estimated Maximal Lyapunov Exponent of environments sampled from the *DeepMind Control Suite* when controlled by SAC, TD3 and Dreamer V3 (DR3). Each policy-environment combination is independently trained with three random seeds and evaluated using the parameters in Table 5, except the number of iterations, which varies from 1 to 100. The mean and 95% confidence interval indicate that MLE converges after 100 iterations.

## B.3 Number of samples ablation study

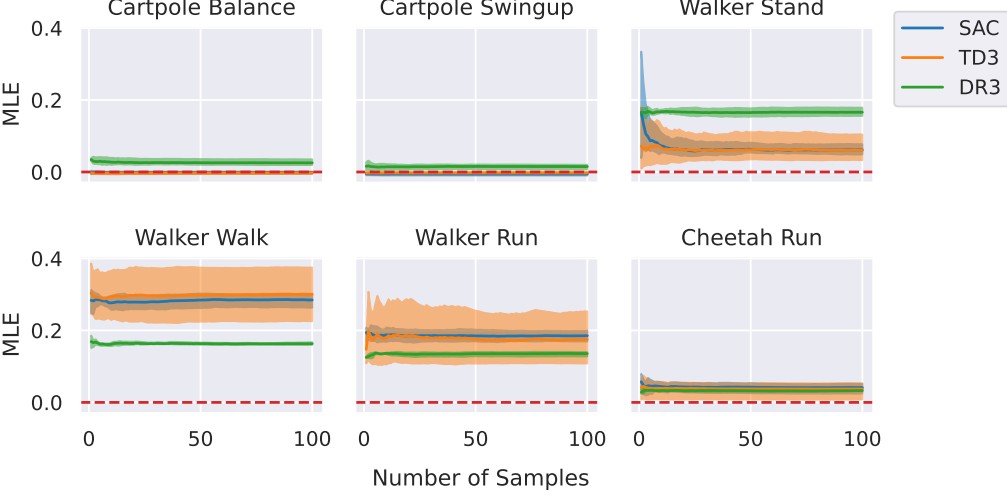

Figure 10: Estimated Maximal Lyapunov Exponent of environments sampled from the *DeepMind Control Suite* when controlled by SAC, TD3 and Dreamer V3 (DR3). Each policy-environment combination is independently trained with three random seeds and evaluated using the parameters in Table 5, except the number of initial samples, which varies from 1 to 20. The mean and 95% confidence interval indicate that MLE converges with 20 initial samples.

