# OpenReview forum: "Enhancing Robustness in Deep Reinforcement Learning: A Lyapunov Exponent Approach"
_NeurIPS.cc/2024/Conference — NeurIPS 2024 poster_

### Official Review · Reviewer_2W37 · 2024-06-23

**Soundness:** 4
**Presentation:** 3
**Contribution:** 3
**Rating:** 7
**Confidence:** 4

**Summary:**

This work studies adversarial RL and policy stability from the perspective of the Lyapunov spectrum. A regularization term is introduced to encourage more robust policies.

**Strengths:**

- A novel idea being introduced from classical literature and implemented in the deep RL setting
- The regularization introduced is effective at smoothing out trajectories generated by learned policies
- This can have important real-world effects in safety-critical / sensitive applications
- Opens work for future (theory/computational/applications) studies to build off

**Weaknesses:**

The weaknesses I've found are mostly minor, but I still hope you can address:

- The correlation between Figure 2 and 3 is not immediately clear to me; can you please discuss this further?
- On that note, the improvement in terms of reward is not clear (I know this is not necessarily "the point" but given the discussion of reward trajectories, it would be nice to see further iumprovemetn here
- How error-prone is the calculation of LEs discussed in Sec 2.2?
- It's not very clear to me the use of "no actions" in the figures. Can you explain how this "benchmark" should be interpreted? How about in environments where there is no such "do nothing" action?
- Can you include non-scalar comparisons of algorithms (cf. https://github.com/google-research/rliable)
- The appendix is a bit sparse. Can you provide any further experiment details?

**Questions:**

- Can you please provide legends in the figures? If you have the data, can you include a third panel in Fig. 6 corresponding to the regularized policy? (otherwise this figure is ambiguous without comparison)
- Maybe I missed it, but what is $S$ in Eq. 4?
- Can you  measure the divergence in trajectories such as in Fig. 4 & 8 and compare to the calculated LE?
- By how much do you weight the regularization in Eq. 5?
- Can you discuss the use of hidden states $h$? It isn't clear to me how this affects the loss beyond what the first term contributes

These questions may open new areas of research, and I don't intend for them to be answered within this paper, but I'm curious to hear your thoughts on the following:
- How does this work relate to the idea of imposing Lipschitz constraints / regularization on DNNs? I see it as an orthogonal direction, but can they solve the same problem? If so, can they be compared?
- Can we use pre-trained policies, then finetune them wrt the LE regularization to make them more robust?
- In 2.3, can you discuss the case of $\gamma \to 1$?

minor typographical:
- Eq. 1, I believe you need a conditional on the state $s$ and an expectation over initial states.
- L161: "...low levels of chaos **as** they have..."
- Eq. 5 use parentheses over both t-dep terms
- Please increase fontsizes in Fig 7

**Limitations:**

Can you please include limitations in Sec 7?

---

> ### Author Rebuttal · Authors · 2024-08-07
>
> Thank you very much for taking the time to review our work. We are delighted that you are interested in our paper! The question you raised is very insightful, and we would like to share some opinions about it.
> ### Weaknesses
> - **The correlation between Figure 2 and 3 is not immediately clear to me**\
> These figures show three different metrics attained by various deep RL agents when controlling environments sampled from the Deep Mind control suite. Figure 2 provides the average reward attained, while Figure 3 provides the estimated MLE and SLE for each policy-environment pair. We include these figures to demonstrate that control systems which appear to be performing well can still produce chaotic and unstable state dynamics.
> - **The improvement in terms of reward is not clear**\
> Improving the performance of Dreamer V3 is not the objective of this work, as we instead focus on improving the stability of the state trajectories. We expect that including the regularisation term introduces a tradeoff between stability and performance. Despite this, we find that regularised policies perform better than the state-of-the-art approach in four out of the seven test environments. Furthermore, when noise observation noise is introduced, we find our approach matches or outperforms all baselines in all the environments.
> - **How error-prone is the calculation of LEs discussed in Sec 2.2?**\
> When calculating the LEs using the method outlined by Benettin et al., we need to define the initial perturbation size, time horizon, normalisation frequency, and the number of initial states. We ran ablation studies for all these values, which can be included in the Appendix of the camera-ready version. From these additional experiments, we find that the calculation of the LEs is consistent for all parameter values except initial perturbation size, which can cause differing LE values when too large. Therefore, the calculation of the LEs discussed in Section 2.2 is not error-prone, providing a suitably small initial perturbation size.
> - **It's not very clear to me the use of "no actions" in the figures.**\
> The "no action" benchmark is used to show that chaos is introduced by the DNN policy and not the environment. At each time step, this baseline applies the fixed action $\textbf{0}^m$; thus, no additional torques are applied to the available joints. This baseline is used to evaluate the stability of each environment without any control intervention. Since each environment has MLE = 0 and SLE $\leq$ 0 when controlled by the "no action" baseline, we find that these systems are naturally invariant to small perturbations. In contrast, when controlled by deep RL policies, these environments have nonzero MLE. Therefore, we can conclude that the level of chaos found in the control interaction is produced by the DNN policies.
> - **Can you include non-scalar comparisons of algorithms?**\
> This will be included in the camera-ready version.
> - **The appendix is a bit sparse. Can you provide any further experiment details?**\
> Further experiment details, including Lyapunov Exponent ablation studies, will be included in the appendix of the camera-ready version.
> ### Questions
> - **Can you please provide legends in the figures? If you have the data, can you include a third panel in Fig. 6 corresponding to the regularized policy?**\
> This will be included in the camera-ready version.
> - **Maybe I missed it, but what is $S$ in Eq. 4?**\
> $S$ is a normalising term used to approximately scale the returns to the range [0,1]. For further details of Equation 4, please refer to the comment in the Global Response.
> - **Can you measure the divergence in trajectories such as in Fig. 4 & 8 and compare to the calculated LE?**\
> These values will be included in the camera-ready version.
> - **By how much do you weight the regularization in Eq. 5?**\
> The regularisation term used in Equation 5 is weighted equal to the policy loss term. We are currently investigating the impact varying this weight has on the balance between performance and stability.
> - **Can you discuss the use of hidden states $h$?**\
> When improving the stability of Dreamer V3, it is important to constrain the internal hidden states, as the policy network uses this and an embedding of the current observation to produce an action. Therefore, simply constraining the prediction observations can still create chaotic dynamics, as differing internal representations can produce different long-term outcomes.
> - **How does this work relate to the idea of imposing Lipschitz constraints/regularization on DNNs?**\
> Improving the smoothness of DNNs by imposing Lipschitz constraints is similar to our problem, as they both consider the smoothness of a function with respect to its inputs. However, when determining the level of chaos, the function we consider is the repeated composition of the dynamical system transition function. Due to this repeated composition, a continuously differentiable function, and thereby Lipschitz continuous, can produce chaotic dynamics (e.g. The Lorenz Attractor). Therefore, imposing Lipschitz constraints/regularisation on the DNN policy does not guarantee stable long-term dynamics.
> - **Can we use pre-trained policies, then finetune them wrt the LE regularization to make them more robust?**\
> This is an active area of research which we are currently investigating.
> - **In 2.3, can you discuss the case of $\gamma \rightarrow 1$?**\
> As $\gamma \rightarrow 1$, the objective function becomes Lipschitz continuous for control systems with $\lambda_1 < 0$ and non-differentiable for systems with $ \lambda_1 > 0 $. For further details, please refer to the original paper by Wang et al.
> - **Typographical Errors**\
> These errors will be fixed for the camera-ready version.
> - **Limitations**\
> The limitations of our work have been discussed in Section 6; however, for clarity, we will include a limitations section in the camera-ready version.

---

> > ### Comment · Reviewer_2W37 · 2024-08-12
> >
> > I appreciate the authors' responses, thank you for addressing the questions raised. I believe the revised version will make for a great paper. I will maintain the current score as it appears the reviewers have converged.

---

### Official Review · Reviewer_pxFr · 2024-07-02

**Soundness:** 3
**Presentation:** 2
**Contribution:** 2
**Rating:** 6
**Confidence:** 4

**Summary:**

Deep RL methods are usually lack of robustness in control tasks whose dynamics are chaotic, thereby having positive maximal Lyapunov exponents (MLEs). This paper proposes an approach that improves the stability of trained deep RL controller through MLE regularization.

**Strengths:**

* The problem studied in this paper is well-motivated and important.

* Characterizing the robustness using maximal Lyapunov exponents is promising.

* The presentation is straightforward.

**Weaknesses:**

* This paper dedicates many pages to the chaotic phenomena in RL (Sections 3 and 4), which have been addressed in previous works as introduced in Sections 1 and 2. The core contribution, which I believe is the regularization technique, needs more elaboration and analysis.

* There is no concrete algorithm provided that the audience can follow to easily implement the approach proposed in this paper. Imagine a person with little or no knowledge of dynamical systems. Can they manage to implement the algorithm after reading the paper?

* (minor) Some information is missing. For example, what are $S$, $H$ and $v_\phi(s_t)$ in equation (4)?

**Questions:**

* Does the policy network architecture affect the performance of your approach?

* Is it possible to find two trajectories with equal rewards, where one is stable and the other is chaotic? In other words, the reward itself may not be sufficient to reflect the stability of a trajectory.

* How does the approach perform in the case that it is unable to estimate the MLE accurately in practice?

**Limitations:**

It has discussed its empirical limitations in Section 6.

---

> ### Author Rebuttal · Authors · 2024-08-07
>
> Thank you for your time and thoughtful review. We value your insightful questions, and you can find our response below.
>
> ### Weaknesses
> - **This paper dedicates many pages to the chaotic phenomena in RL (Sections 3 and 4), which have been addressed in previous works as introduced in Sections 1 and 2.**\
> While the idea of chaos in RL has been addressed in prior works, they focus on the ***sensitivity of the value function subject to policy parameter perturbations during training***. In these works, it is demonstrated that the return surface can have a fractal structure, and this can produce poor policy updates during training. \
> In contrast, the analysis in Sections 3 and 4 focuses on the ***sensitivity of a fixed control system subject to state perturbations after training***. The results in these sections demonstrate that trained state-of-the-art model-based and model-free methods produce a chaotic control interaction in which a small change in the system state can have a profound impact on the long-term state trajectories and the reward attained. As such, these deep-RL methods cannot provide the stability guarantees necessary for real-world control systems.
>
> - **The core contribution, which I believe is the regularization technique, needs more elaboration and analysis.**\
> We appreciate your suggestion for further exploration and study into the impact of MLE regularisation. We would like to highlight that a series of experiments have been conducted using multiple seeds which demonstrate that the inclusion of MLE regularisation significantly improves the stability of the control interaction and often increases the total reward attained (Table 2). Furthermore, this increased stability improves the performance of Dreamer V3 when noise is added to the observation space (Figure 7). Examining the state trajectories produced by Dreamer V3 and Dreamer V3 + MLE regularisation when controlling the *Walker Stand* task (Figure 8) shows the regularisation improves the consistency of the control interaction as the trajectories do not diverge significantly. These results provide strong evidence that MLE regularisation leads to significant improvements in the stability of the state trajectories produced by Dreamer V3.
>
> - **There is no concrete algorithm provided that the audience can follow to easily implement the approach proposed in this paper.**\
> We appreciate your comment regarding the clarity of the proposed regularisation method. We would like to highlight that in Section 5, we outline the additional loss term used and that this can be calculated by estimating the state and hidden-state trajectories. However, we acknowledge this is unclear, so we will include a formal algorithm in the camera-ready version.
>
> - **What are $S$, $H$ and $v_\phi(s_t)$ in equation (4)?**\
> Please, see the comment outlined in the Global Response, in which we address the definition of Equation 4.
>
> ### Questions
> - **Does the policy network architecture affect the performance of your approach?**\
> From our preliminary experiments, we find that the network architecture does not impact stability however, this is an active area of research which we are currently investigating. For our current work, we maintained the same network architecture for the baseline Dreamer V3 and MLE regularised Dreamer V3 to allow for a fair comparison.
>
> - **Is it possible to find two trajectories with equal rewards, where one is stable and the other is chaotic?**\
> Yes, it is possible for two trajectories to attain equal rewards despite having differing levels of stability. Consider the Cartpole Balance task as an example. In this control system, the agent is provided +1 reward per step for maintaining the pole within 1° of the vertical upright position and within 0.25 meters of the centre of the track. As such, a trajectory that maintains a stable vertical position will attain the same reward as a chaotic trajectory that remains within the high-reward region.
>
> - **How does the approach perform in the case that it is unable to estimate the MLE accurately in practice?**\
> As the estimation of MLE used for the regularisation term uses a learned model of the system dynamics, it is possible for the estimation to be inaccurate. When this incorrect estimation of MLE is included in the policy loss, the updated policy can produce more unstable dynamics. However, avoiding this inaccurate estimation requires an improvement in dynamics prediction models, which is outside the scope of this paper.

---

> > ### Comment · Reviewer_pxFr · 2024-08-09
> > **Reply to rebuttal**
> >
> > I would like to thank the authors for their rebuttal and it has resolved my concerns. I still recommend the authors to add a formal algorithm table for their approach so that the audience can easily implement it. I am raising my score to 6.

---

> > > ### Author Response · Authors · 2024-08-09
> > >
> > > Thank you very much for your positive feedback and for raising your score. We are delighted to hear that we have effectively addressed your concerns and appreciate your suggestion to add a formal algorithm. We agree that it will enhance the clarity of our work and will, therefore, incorporate this into our camera-ready version.

---

### Official Review · Reviewer_TDUf · 2024-07-12

**Soundness:** 3
**Presentation:** 3
**Contribution:** 3
**Rating:** 6
**Confidence:** 4

**Summary:**

To address the issue of stability of deep reinforcement learning, the authors first gauge the chaotic behavior of various state-of-the-art deep reinforcement learning policies in continuous control environments, and quantify the stability of those policies with significant impact of their applicability to real world problems. Then, the authors proposed an improvement based on implementing a Maximal Lyapunov Exponent regularization in the RL architecture, and demonstrate the improvement with examples.

**Strengths:**

Lyapunov exponent is an important concept developed in dynamical system. There are many related sophisticated research conducted. It is beneficial to both machine learning and dynamical system to have used it in the analysis of deep RL.

Sec 3 & 4 present a reasonable analysis on the stability study of deep RL.

**Weaknesses:**

The proposed method on maximal Lyapunov exponent regularization needs to be further explored and studied, both in terms of the theory involved and amount of experiments conducted. The idea is novel, but not fully explored and explained. The current set of numerical experiments are limited and not truly convincing.

**Questions:**

Could more explanation of all the terms and parameters in Equation (4) be provided? Also, how is the proposed regularizing term in (5) is related to (4)?

**Limitations:**

The authors adequately addressed the limitations.

---

> ### Author Rebuttal · Authors · 2024-08-07
>
> We greatly appreciate your time and effort in reviewing our work. Your questions are very insightful, and we would like to offer our thoughts on it.
>
> ### Weaknesses
> - **The proposed method on maximal Lyapunov exponent regularization needs to be further explored and studied, both in terms of the theory involved and amount of experiments conducted.** \
> We appreciate your suggestion for further exploration and study into the impact of MLE regularisation. We would like to highlight that we have outlined the theory used for the regularisation method in Section 5 and have provided extensive experimental results using multiple seeds in Section 6. These results use the same environments as the original Dreamer V3 paper and demonstrate that the inclusion of MLE regularisation improves the stability of the control interaction and often the total reward attained (Table 2). We acknowledge that extending these experiments to more complex control systems is desirable. However, we believe that the results outlined in our work sufficiently demonstrate the benefits of MLE regularisation.
>
> ### Questions
>
> - **Could more explanation of all the terms and parameters in Equation (4) be provided?** \
> Please, see the comment outlined in the Global Response, in which we address the definition of Equation 4.
>
> - **How is the proposed regularizing term in (5) related to (4)?** \
> The loss function used to train the MLE regularised Dreamer V3 policy is $\mathcal{L}^\text{Policy}(\theta) + \mathcal{L}^{\lambda_1}(\theta)$. This will be clarified in the camera-ready version.

---

> > ### Comment · Reviewer_TDUf · 2024-08-10
> > **Response to the rebuttal**
> >
> > Thank you the the rebuttal. They address some of my concerns. I would also like to raise my score to (6).

---

### Author Rebuttal · Authors · 2024-08-07

We would like to express our gratitude to all the reviewers for taking the time to review our paper and providing valuable feedback. Your comments have been insightful and have certainly contributed to the refinement of our work. We appreciate the effort you have put into the review process. After carefully considering your comments, we would like to address the general concerns and criticisms raised during the review.

- **Could more explanation of all the terms and parameters in Equation 4 be provided?** \
Equation 4 outlines the loss function used by Hafner et al 2024 to train Dreamer V3’s policy network. All the terms used are consistent with the original article, including Stop Gradient (sg), Return Estimates ($R^\lambda_t$), Critic Estimates ($v_\phi(s_t)$), Scaling Factor ($S$), Entropy Scale ($\eta$) and Entropy Regulairsor (H$[\cdot]$). This equation is included in Section 5 to provide background information for the reader as the MLE regularisation term (Equation 5) is added to this equation. This will be clarified in the camera-ready version.

---

### Decision · Program_Chairs · 2024-09-25

**Decision:**

Accept (poster)

**Comment:**

This paper studied the robustness of deep RL policies to a single small state perturbation in deterministic continuous control tasks. It first showed the vulnerability of deep RL policies, as being deterministically chaotic against small perturbations, which further indicates two facets of chaotic behaviors of the policies in real-world applications. To address the issue, the paper proposed a novel regularisation technique based on Maximal Lyapunov Exponent. The new algorithm reduces the chaotic behaviors and made the learned policies more resilient, with extensive experimental results. It reaches a consensus that the observations in this paper are new and of great importance to deep RL, and the proposed workaround is novel. Recommend acceptance. I suggest the authors to address the remaining comments from the reviewers in preparing the final version of the paper.